# Significance of Pelvic Fluid Observed during Ovarian Cancer Screening with Transvaginal Sonogram

**DOI:** 10.3390/diagnostics12010144

**Published:** 2022-01-07

**Authors:** Justin W. Gorski, Charles S. Dietrich, Caeli Davis, Lindsay Erol, Hayley Dietrich, Nicholas J. Per, Emily Lenk Ferrell, Anthony B. McDowell, McKayla J. Riggs, Megan L. Hutchcraft, Lauren A. Baldwin-Branch, Rachel W. Miller, Christopher P. DeSimone, Holly H. Gallion, Frederick R. Ueland, John R. van Nagell, Edward J. Pavlik

**Affiliations:** 1Division of Gynecologic Oncology, University of Kentucky Markey Cancer Center, Lexington, KY 40536, USA; justin.gorski@uky.edu (J.W.G.); charles.dietrich@uky.edu (C.S.D.III); anmcdowe@gmail.com (A.B.M.); mbri229@uky.edu (M.J.R.); Megan.hutchcraft@uky.edu (M.L.H.); labald1@uky.edu (L.A.B.-B.); raware00@uky.edu (R.W.M.); christopher.desimone@uky.edu (C.P.D.); holly.gallion@uky.edu (H.H.G.); fuela0@uky.edu (F.R.U.); jrvann2@uky.edu (J.R.v.N.J.); 2Denison University, Granville, OH 43023, USA; daviscaeli@gmail.com; 3Tripler Army Medical Center, Honolulu, HI 96859, USA; lindsay.a.erol.mil@mail.mil; 4Kenyon College, Gambier, OH 43022, USA; dietrich1@kenyon.edu; 5Department of Obstetrics & Gynecology, University of Kentucky, Lexington, KY 40536, USA; nicholas_per@trihealth.com (N.J.P.); emily.lenk@uky.edu (E.L.F.)

**Keywords:** transvaginal ultrasound, ovarian cancer screening, pelvic fluid, abdominal fluid, free fluid

## Abstract

The primary objective was to examine the role of pelvic fluid observed during transvaginal ultrasonography (TVS) in identifying ovarian malignancy. A single-institution, observational study was conducted within the University of Kentucky Ovarian Cancer Screening trial from January 1987 to September 2019. We analyzed true-positive (TP), false-positive (FP), true-negative (TN), and false-negative (FN) groups for the presence of pelvic fluid during screening encounters. Measured outcomes were the presence and duration of fluid over successive screening encounters. Of the 48,925 women surveyed, 2001 (4.1%) had pelvic fluid present during a TVS exam. The odds ratio (OR) of detecting fluid in the comparison group (TN screen; OR = 1) significantly differed from that of the FP cases (benign pathology; OR: 13.4; 95% confidence interval (CI): 9.1–19.8), the TP cases with a low malignant potential (LMP; OR: 28; 95% CI: 26.5–29.5), TP ovarian cancer cases (OR: 50.4; 95% CI: 27.2–93.2), and FN ovarian cancer cases (OR: 59.3; 95% CI: 19.7–178.1). The mean duration that pelvic fluid was present for women with TN screens was 2.2 ± 0.05 encounters, lasting 38.7 ± 1.3 months. In an asymptomatic screening population, free fluid identified in TVS exams was more associated with ovarian malignancy than in the control group or benign ovarian tumors. While pelvic free fluid may not solely discriminate malignancy from non-malignancy, it appears to be clinically relevant and warrants thoughtful consideration.

## 1. Introduction

Ovarian cancer continues to be the most lethal gynecological cancer with most patients facing a diagnosis of late stage metastatic disease that is associated with a five-year survival of only 30%. While the lifetime risk of ovarian cancer is less than breast cancer, it has a death-to-incidence ratio that is three to four times more than breast cancer [1,2,3]. Ovarian cancer can arise from the ovary as well as the fallopian tube [4,5,6,7,8,9,10,11] and occur as five major histological subtypes (high-grade serous, low-grade serous, endometrioid, clear cell, and mucinous) that have been explored by immunochemical, genetic, and homologous recombination approaches [12]. Deleterious mutations in DNA repair genes can drive defective homologous recombination and are emerging biomarkers of sensitivity/insensitivity to poly (ADP-ribose) polymerase (PARP) inhibitors which interfere with the ability of PARP to repair treatment-mediated DNA damage in cancer cells [13]. Despite technically advanced treatment strategies in precision medicine and immunotherapy, long-term durable treatment responses have not yet been achievable [14,15]. However, long-term survival rates that are greater than 90% in women with stage I ovarian cancer have focused on an advocacy for screening efforts to detect early-stage disease [16].

Transvaginal ultrasonography (TVS) is a safe, minimally invasive [17] and cost-effective [18,19] modality that allows the visualization of echogenic structures and non-echogenic fluid. TVS has been used as a major modality in four major screening trials for ovarian cancer including the Kentucky Ovarian Screening (KYOVS) trial [18,20], the Prostate, Lung, Colorectal, and Ovarian (PLCO) Cancer Screening Randomized Controlled Trial [21], the UK Collaborative Trial of Ovarian Cancer Screening (UKCTOCS) [22,23], and the Shizuoka Cohort Study of Ovarian Cancer Screening (SCSOCS) trial [24]. The reported findings from these studies are summarized in Table 1. Routine TVS can identify free fluid in the abdomen or pelvis. Of the four major ovarian cancer screening trials that utilized TVS, none examined the degree to which free fluid is associated with malignancy. Consequently, there are no clear guidelines on the recommended management of free fluid incidentally identified during a gynecologic ultrasound. Historically, physicians performed culdocentesis to sample fluid in the pouch of Douglas [25], but this has been replaced by ultrasound-guided paracentesis. Clinical guidelines are even less clear on how to manage small-volume pelvic fluid collections.

In this report, we examined the frequency and duration of free fluid during TVS exams in a large screening population and correlated the findings with the patient diagnosis. Our primary objective was to determine if free fluid provides additional information in predicting ovarian malignancy compared to ultrasound alone.

## 2. Materials and Methods

### 2.1. Subjects

Women enrolled in the KYOVS trial from January 1987 to September 2019 were eligible for study inclusion. The University of Kentucky institutional review board for human studies approved this prospective cohort trial (Transvaginal Ultrasonography as a Screening Method for Ovarian Cancer, IRB #45030, renewed on 3 December 2021). Eligibility criteria included asymptomatic women aged ≥50 years and asymptomatic women aged ≥25 years with a documented family history of ovarian cancer in at least one primary or secondary relative. Excluded from enrollment were women with a known ovarian tumor or a personal medical history of ovarian cancer. All study patients provided written informed consent before undergoing screening with transvaginal ultrasound. A board-certified gynecologic oncologist reviewed all ultrasound images. The follow-up algorithm of normal and abnormal scans has been previously published [18].

### 2.2. Interpretation of TVS Screening Results

Individuals were designated as a true-positive (TP), if they had histologically confirmed primary epithelial ovarian cancer, a low malignant potential ovarian tumor, a non-epithelial ovarian malignancy, or metastatic disease to the ovary. A designation of a true-negative (TN) screen was applied when a diagnosis of ovarian cancer did not occur for at least 12 months after a “normal” TVS exam. False-positive (FP) screens included an abnormal TVS screen found surgically to have benign ovarian histology, including serous cystadenoma, endometrioma, mucinous cystadenoma, cystic teratoma, fibroma, thecoma, Brenner tumor, leiomyoma, hydrosalpinx, paratubal cyst, and hemorrhagic cyst. We categorized study patients with a diagnosis of ovarian cancer occurring within 12 months of a screening TVS exam that reported “normal” findings as a false-negative (FN).

### 2.3. Pelvic Free Fluid Identification

All sonographers in the University of Kentucky Ovarian Cancer Screening Program are certified by the American Registry for Diagnostic Medical Sonography (ARDMS, Rockville, MD, USA). We digitally recorded the findings of each ultrasound, including the presence or absence of pelvic fluid, in case report forms maintained on a Medlog database (Medlog Systems, Incline Village, NV, USA). We interpreted any amount of free fluid as a positive finding, including the trace free fluid, small volume fluid (≤10 mL), and fluid collections exceeding 10 mL. We determined the overall rate of free fluid identification as the ratio of fluid positive cases to the total number of cases in individual categories.

### 2.4. Statistical Analysis

We compared demographic characteristics using Χ^2^ or Fisher’s exact test for categorical data. We assessed continuous variables for normality and used non-paired t-tests or Wilcoxon rank-sum tests as appropriate. For all analyses, we used a two-tailed test with a significance level of *p* < 0.05. We used GraphPad Prism 5.01 (GraphPad Software, San Diego, CA, USA) to perform the analyses.

We calculated the probability ratio (PR) for each group as the probability of an event in one of the screening groups (TP, TN, FP, and FN). For example, the screening groups consisted of women with fluid and without fluid relative to women with and without fluid in the comparison group. The comparison group consisted of premenopausal normal women (TN) with a body mass index (BMI) of <30. Pre- and post-menopausal women with a BMI of ≥30 were found to have a lower PR of free fluid than those with a BMI of <30; however, a very large body habitus can limit the detection of free fluid by TVS. For this reason, we defined the comparison group as subjects with a BMI of <30. We used VassarStat based on logistic regression to calculate PRs, odds, odds ratios (ORs), log odds, Phi coefficients of association, association identified by Χ^2^-square tests, Fisher’s exact probability, and confidence intervals (CIs) [26].

### 2.5. Fluid Duration Analysis

The duration of free fluid was expressed both as the number of consecutive TVS exams during which fluid was present and as a time duration during consecutive TVS exams.

## 3. Results

Since 1987, the University of Kentucky Ovarian Screening Program has performed 326,998 TVS screens on 48,925 women. We observed free fluid in 2001 (4.1%) of those encounters. TP screens included 78 ovarian malignancies (13 fluid-positive, 16.7%), 20 tumors of a low malignant potential (two fluid-positive, 10%), and 23 malignancies of a non-ovarian origin (three fluid-positive, 13%). Only one of the TP cases was observed to have fluid present in a prior normal TVS exam. There were 614 FP screens classified as high risk for ovarian cancer but found to have benign pathology (31 fluid-positive, 5%). Nine of these cases occurred in a prior normal TVS examination. The TN screens included 41,996 cases that screened negative for malignancy and did not develop ovarian carcinoma (1948 fluid-positive (4.6%): 1071 fluid-positive cases were associated with a normal TVS examination, while 877 were associated with an abnormal TVS examination). There were 21 cases of ovarian cancer diagnosed within 12 months of an FN TVS scan (four fluid-positive, 19%). All of these cases occurred in the absence of an abnormal TVS exam.

### 3.1. Demographics

The demographic characteristics of the TN cohort demonstrated some significant differences between fluid-positive and -negative subgroups. The fluid-negative subgroup was older and had a higher parity and a higher BMI. They were also less likely to be nulliparous or have a family history of ovarian or colon cancer. Although they were less likely to have ever used hormone replacement therapy (HRT), more were using HRT at the time of their last visits (Table 2).

The fluid-positive and fluid-negative subgroups with a TP (Table 3) or FP screen (Table 4) were similar, except for a higher parity in the FP fluid-negative group (*p* = 0.0344, Table 4). We omitted the FN screen demographics because of the small sample size.

### 3.2. Probability of Identifying Pelvic Fluid in Each Group

The PRs and the ORs were determined for women designated as TN for ovarian cancer based on TVS findings. These women were then grouped on the basis of the menopausal status, their BMI, and normal vs. abnormal TVS exams (Table 5).

Using this categorical basis to normalize the PR and OR determination, we observed that TN premenopausal women with a BMI ≥30 had the lowest PR and OR of fluid observed during TVS exams. However, it was possible that a large body habitus might have interfered with detecting fluid in these women during their TVS exams. This effect would be least likely in TN premenopausal women with a BMI <30 that had received a normal TVS exam, and we selected this group to normalize the PRs and the ORs. In this analysis, the PRs ranged from 0.24 to 11.73, while the ORs ranged from 0.24 to 12.26 (Table 5), so that the entire ranges of the PRs and the ORs were defined for every TN category. Next, we used the same approach to compare the PRs and the ORs for women with positive TVS findings with regard to fluid status (Table 6).

Findings associated with ovarian malignancy were statistically different from non-malignant cases (PR_OvCa_ = 36.14–40.48 vs. PR_Benign_ = 1–12.16; *p* < 0.001). There was no statistical difference in the detection of free fluid between TP women with a low malignant potential (LMP; PR: 18.07; 95% CI: 4.71–69.3) and TP women with ovarian cancer (PR: 36.14; 95% CI: 21.36–61.14) or women with benign findings (PR: 12.16; 95% CI: 8.35–17.7) (Figure 1).

The PRs for women with benign findings (identified by red font “1”) or women with tumors of an LMP (identified by red font “2”) were not significantly different (*p* > 0.05) from women who were TN for ovarian cancer (Figure 1, black data points). The PR for fluid in women who were TP for ovarian cancer (Figure 1 identified by red font “3”) was much higher and statistically different (*p* < 0.001) from any category of women who were TN for ovarian cancer (Figure 1, black data points). The PR for fluid in women who were designated FN by TVS alone (Figure 1 identified by red font “4”) was also much higher and statistically different (*p* < 0.001) from any category of women who were TN for ovarian cancer while not significantly different (*p* > 0.05) from women with fluid who were TP for ovarian cancer (Figure 1 identified by red font “3”). This observation indicates that the presence of fluid may be an indicator of malignancy that should be followed further by serial TVS of shorter time intervals.

### 3.3. Duration of Fluid Identified in TN Subjects

Out of the 1948 TN cases, free fluid was detected in a single encounter for 1106 women (56.7%); however, fluid persisted beyond three encounters for only 9% of the study group. On average, fluid persisted for a mean of 2.2 TVS exams (SEM: standard error of the mean = ±0.05 TVS exams). Notably, 28 TN women (1.4%) had persistent fluid for over 10 encounters (Figure 2). The majority of fluid volumes were small, with only 17.5% of TN volumes larger than 10 mL.

In TN cases where pelvic fluid was present for more than one TVS exam (*n* = 843), fluid persisted for a mean of 38.7 months (SEM: ± 1.3 months) (Figure 3). The median duration until fluid resolution in the subset of subjects with two or more encounters was 23.9 months, indicating a skew to the left for the distribution of duration.

Overall, fluid resolved in 77% of TN cases with no statistical difference between persisting fluid-positive cases for normal (2.21 ± 0.07) versus abnormal (2.08 ± 0.07) encounters.

## 4. Discussion

Abdominopelvic ascites is a well-known sign of advanced ovarian, fallopian tube, or peritoneal malignancy, but the importance of free fluid during screening or diagnostic ovarian TVS is less evident. The findings of our investigation confirmed an association between free fluid identified on ovarian TVS and ovarian malignancy. To predict ovarian malignancy, the ADNEX model (Assessment of Different NEoplasias in the adneXa) created by the European International Ovarian Tumour Analysis study group considers nine variables in a polytomous prediction to categorize malignant risk [27,28]. The ADNEX model does not include pelvic fluid as a variable, nor does the Risk of Malignancy Index, or the Risk of Ovarian Malignancy Algorithm [29,30,31,32,33,34]. The University of Kentucky Morphology Index acknowledges fluid as a risk factor, particularly when associated with complex tumors [35]. Our study results support this conviction, as the presence of free fluid was highly associated with the diagnosis of ovarian malignancy (PR_TP_ = 36.14; OR_FN_ = 40.48). This is in keeping with the evidence that potentially malignant ovarian tumors become morphologically more complex as they progress toward malignancy over time [36]. Rarely, pelvic fluid may be the only early sonographic feature of gynecologic malignancy. Including free fluid as a risk variable for ovarian malignancy has the potential to improve detection sensitivity by reducing FN results.

The results reported here indicated that the presence of fluid is highly probable in cases of frank ovarian malignancy and less so in cases of benign findings and LMP tumors. The authors appreciate that free fluid is present in many benign conditions, including hemorrhagic cysts, fibromas, endometriomas, and ruptured serous or mucinous cystadenomas. However, our results showed a statistically less frequent association of free fluid in benign tumors versus ovarian malignancy (PR = 12.16 vs. PR = 36.14; *p* < 0.05). For non-malignant cases, more than half (56.7%) of the TN screens had fluid present for one encounter, 9% for more than three, and 1.4% for more than 10 encounters. Thus, our data indicates that free fluid mostly disappears in non-malignant scenarios if serially monitored. Because of this observation, it can be asked if fluid might be due to follicular fluid entering the peritoneal cavity during ovulation. This possibility is unlikely, because premenopausal TN screens were similarly low in the observation of fluid as postmenopausal cases (Figure 1 A–L). In addition, the six TP cases that were premenopausal at diagnosis were negative for fluid. We recommend serial surveillance with TVS, until a malignant morphology is suspected or the fluid resolves. Speculation on the source of fluid associated with malignant cases could include the leakage of cyst fluid from ovarian cysts containing malignancy as an explanation for fluid observed in the adnexa.

Only a few studies have reported the presence of pelvic fluid in investigations of ovarian cancer. An underpowered study of 17 women evaluated an intra-ovarian blood flow with 3-dimensional power Doppler ultrasound before surgical intervention in women suspected of having ovarian carcinoma [37] and found no difference in pelvic fluid for women with (16.7%) or without ovarian malignancy (18.2%; *p* = 0.938). Another study of 60 women with histologically confirmed primary peritoneal cancer investigated disseminated peritoneal carcinomatosis using TVS and documented the presence of free pelvic fluid [38]. Given the advanced stage of disease at inclusion, it is not unexpected that the study found 83% of women had free fluid present in the pelvic cul-de-sac. Lastly, investigators studied free pelvic fluid with magnetic resonance imaging (MRI) in women with ovarian tumors. Eighty-seven women underwent surgery for pelvic tumors of unknown malignant potential. Preoperative MRI indicated that the presence of large peritoneal fluid pockets is moderately predictive of malignancy or peritoneal spread of the tumor [39]. Despite this, the recent American College of Radiology Ovarian-Adnexal Reporting and Data System did not include fluid as a variable in the scoring algorithm for sonographically indeterminate adnexal masses [40,41].

This study has several strengths. These data resulted from a substantial cohort of patients all screened in one program where operations and techniques were well standardized. Throughout this study, the same Director of Research, Edward J. Pavlik (PhD) provided consistent oversight and the same clinician, John R. VanNagell Jr., (MD) reviewed all TVS scans. Lastly, it is relatively easy to identify free fluid on pelvic ultrasound because of the inherent contrast between water-dense fluid and pelvic soft tissue.

There are also some study limitations. First, sonographers did not quantify the volume of the pelvic free fluid on all studies. As a result, we included those with recorded fluid volumes and test entries indicating fluid, possible fluid, ascites, and possible ascites. This approach may have underestimated the number of subjects who had pelvic fluid present, because it was not a required outcome measure. Since the peritoneal cavity contains typically 50–75 mL of physiologic fluid that serves to lubricate the abdominal wall and viscera [42,43,44], some fluid-positive cases may be a result of the pooling of this physiologic fluid. In addition, because the views obtained in this screening study were limited to the ovaries, it is possible that abdominal fluid above the pelvic brim or less likely fluid in the pouch of Douglas may not have been observed, especially since the size of the uterus varied in each individual [45]. Lastly, some demographic variables showed small statistical differences between TN fluid-positive and -negative subgroups contributing to potential bias.

## 5. Conclusions

Pelvic fluid identified during TVS screening is associated with ovarian malignancy and should not be disregarded. The presence of free fluid during TVS should heighten awareness for malignancy, particularly for an ovarian tumor with a high-risk morphology. When free fluid associated with benign ovarian tumors is present for more than one encounter, half of the cases resolve within 24 months. Since free fluid is associated with ovarian malignancy, we recommend serial ultrasound surveillance, until a malignant morphology is suspected or the fluid resolves.

## Figures and Tables

**Figure 1 diagnostics-12-00144-f001:**
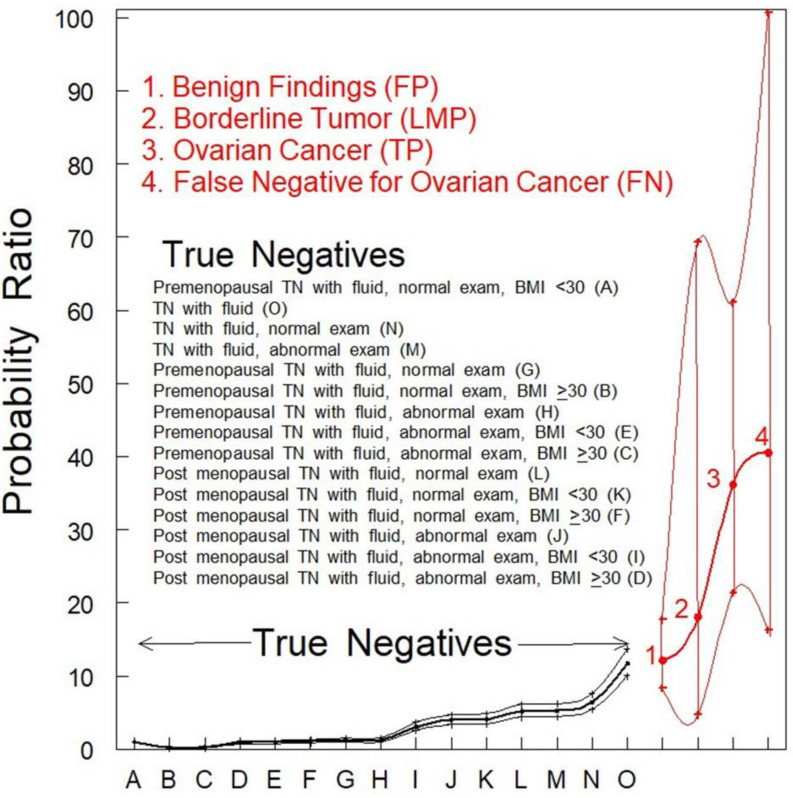
PR analysis of women with pelvic fluid present in TVS. PRs with 95% CIs are shown. TN and benign findings vs. ovarian cancers (OvCa) were significantly different (*p* < 0.05). The classifications of women who were TN for ovarian cancer were rank-ordered as identified by uppercase letters within parentheses inside the figure.

**Figure 2 diagnostics-12-00144-f002:**
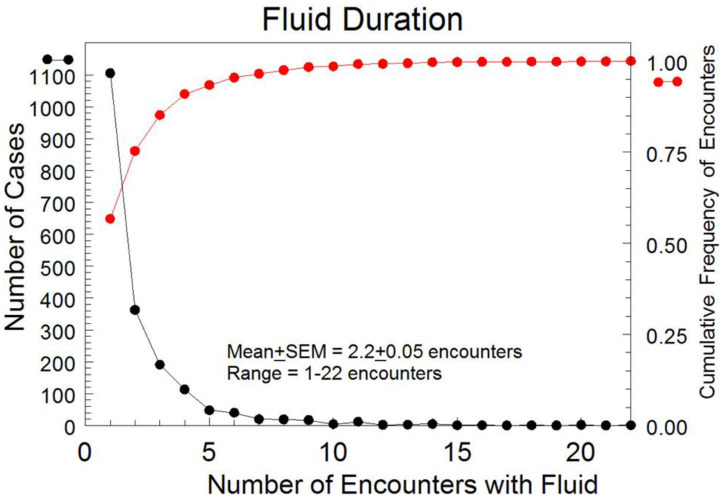
Durations of the detected pelvic free fluid by encounter for all TN cases with free fluid present. **•** represents the cumulative frequency of encounters. **•** indicates the number of cases.

**Figure 3 diagnostics-12-00144-f003:**
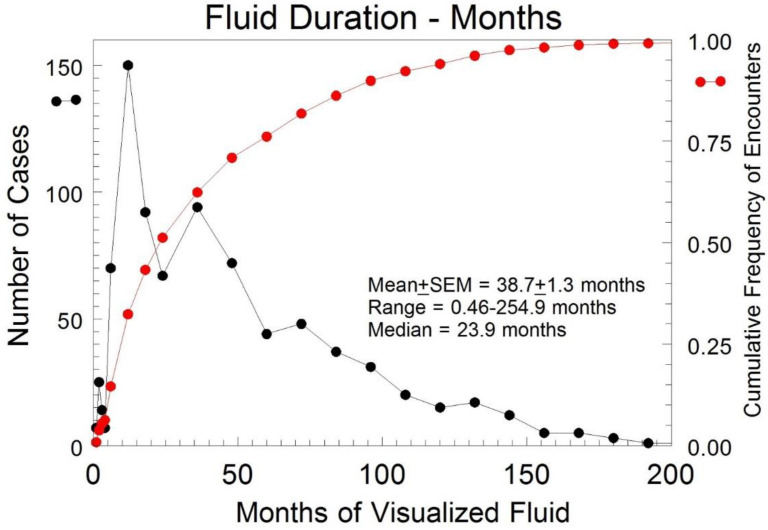
Duration of the detected pelvic fluid in months for all TN cases with free fluid present for at least two encounters. **•** represents the cumulative frequency of encounters. **•** indicates the number of cases.

**Table 1 diagnostics-12-00144-t001:** Summary of ovarian screening trials.

Study	KYOVS	PLCO	SCSOCS	UKCTOCS
Study design	Prospective cohort (ongoing)	Intent to treat RCT (closed)	Intent to treat RCT (closed)	Intent to treat RCT (closed)
Number screened	48,925 ^a^	34,253 ^b^ 34,304 *	41,688 ^b^ 40,799 *	50,625 ^c^ 50,623 ^a^ 101,314 *
Total screens	326,998	150,598	156,747	345,570 ^c^ 327,775 ^a^
Invasive ovarian cancers detected	78	212 ^b^	27	522 ^c^ 517 ^a^ 1016 *
Shift to early-stage disease ^d^	Yes (63%)	No	Yes (67%)	Yes (39.2%)
Survival benefit	Yes	No	No	No ^e^

^a^ US (transvaginal ultrasound) alone; ^b^ US alone followed by Ca125 and Ca125 alone; ^c^ Ca125 followed by US; * control; ^d^ stages I and II; Randomized Control Trial (RCT); ^e^ Yes for incident cases, but under-powered.

**Table 2 diagnostics-12-00144-t002:** Demographic characteristics of true-negative (TN) screen subjects at the first ultrasound.

Demographic Variable	All TN Subjects (*n* = 48,212)	TN and Fluid-Negative (*n* = 46,263)	TN and Fluid-Positive (*n* = 1949)	*p*
Age (y)	57.0, 57 (24–95)	57.3, 57 (24–95)	51.4, 52 (25–91)	<0.0001
Parity	2.3, 2 (0–19)	2.3, 2 (0–19)	2.1, 2 (0–8)	<0.0001
Weight (kg)	73.4, 70.3 (34–204)	73.5, 70.3 (34–204)	70.6, 67.1 (41–153)	<0.0001
Height (cm)	163.3, 162.6 (119–198)	163.3, 162.6 (119–198)	164.3, 165 (135–188)	<0.0001
Family cancer history:				
Ovary	11,329 (23.5%)	10,628 (23.0%)	701 (38.9%)	<0.0001
Breast	23,758 (49.6%)	22,797 (49.2%)	961 (49.3%)	0.9790
Colon	13,433 (27.9%)	12,819 (27.7%)	614 (31.5%)	0.0003
No history of hormone replacement therapy	17,587 (36.5%)	16,711 (36.1%)	876 (44.9%)	<0.0001
Hormone replacement on the last visit	5830 (12.1%)	5627 (12.2%)	203 (10.4%)	0.0205
Nulliparous	7020 (14.6%)	6647 (14.4%)	373 (19.1%)	<0.0001

Data are represented as the mean, median, range within parenthesis, and percentage (%).

**Table 3 diagnostics-12-00144-t003:** Demographic characteristics of true-positive (TP) screen subjects at the first ultrasound.

Demographic Variable	All TP Subjects (*n* = 78)	TP and Fluid-Negative (*n* = 64)	TP and Fluid-Positive (*n* = 14)	*p*
Age (y)	65.5, 66 (36–86)	64.9, 66 (36–82)	68.4, 71 (45–85)	0.2285
Parity	2.0, 2 (0–8)	2.1, 2 (0–8)	1.8, 2 (0–5)	0.4818
Weight (kg)	71.4, 69 (44–122)	72.5, 69.7 (44–123)	67.7, 68.6 (52–82)	0.1920
Height (cm)	163.2, 163 (142–179)	163, 163 (152–178)	164.4, 165 (142–178)	0.4329
Family cancer history:				
Ovary	17 (21.7%)	15 (23.4%)	2 (14.2%)	0.7223
Breast	33 (42.3%)	26 (40.6%)	7 (50%)	0.5614
Colon	20 (25.6%)	16 (25%)	4 (28.6%)	0.7464
No history of hormone replacement therapy	59 (75.6%)	48 (75%)	11 (78.6%)	1
Hormone replacement on the last visit	6 (7.7%)	5 (7.8%)	1 (7.1%)	1
Nulliparous	14 (17.0%)	9 (14.1%)	5 (35.7%)	0.1159

Data are represented as the mean, median within parenthesis, range, and percentage (%).

**Table 4 diagnostics-12-00144-t004:** Demographic characteristics of false-positive (FP) screen subjects at the first ultrasound.

Demographic Variable	All FP Subjects (*n* = 614)	FP and Fluid-Negative (*n* = 581)	FP and Fluid-Positive (*n* = 33)	*p*
Age (y)	59.2, 59 (29–85)	59.3, 59 (29–85)	57.1, 59 (36–81)	0.2992
Parity	2.1, 2 (0–10)	2.1, 2 (0–10)	1.6, 2 (0–4)	0.0344
Weight (kg)	74.8, 72.6 (36–167)	75.1, 72.6 (36–167)	70.9, 69.9 (47–98)	0.1632
Height (cm)	164.4, 162.6 (139–181)	164.4, 162.6 (140–181)	164.8, 165 (152–175)	0.8202
Family cancer history:				
Ovary	182 (29.6%)	168 (28.9%)	14 (42.4%)	0.0983
Breast	269 (43.8%)	252 (43.3%)	17 (51.5%)	0.3592
Colon	161 (6.2%)	148 (25.5%)	13 (39.4%)	0.0770
No history of hormone replacement therapy	60 (9.86%)	58 (10%)	2 (6.1%)	0.7612
Hormone replacement on last visit	41 (6.7%)	39 (6.7%)	2 (6.1%)	1
Nulliparous	31 (5%)	29 (5%)	2 (6.1%)	0.6797

Data are represented as the mean, median within parenthesis, range, and percentage (%).

**Table 5 diagnostics-12-00144-t005:** Probability ratios (PRs) and odds ratios (ORs) of fluid observed by transvaginal ultrasonography (TVS) in women with TN results.

Group	Fluid-Positive	Fluid-Negative	PR (95% Confidence Interval (CI))	OR (95% CI)
Premenopausal TN with fluid, a normal exam and a BMI < 30	166	41,830	1	1
TN with fluid	1948	40,048	11.73 (10.02–13.74)	12.26 (10.45–14.37)
TN with fluid and a normal exam	1071	40,925	6.45 (5.48–7.59)	6.59 (5.60–7.77)
TN with fluid and an abnormal exam	877	41,119	5.28 (4.48–6.23)	5.37 (4.55–6.35)
Premenopausal TN with fluid and a normal exam	207	41,789	1.25 (1.02–1.53)	1.25 (1.02–1.53)
Premenopausal TN with fluid, a normal exam, a BMI ≥ 30	40	41,956	0.24 (0.17–0.34)	0.24 (0.17–0.34)
Premenopausal TN with fluid and an abnormal exam	211	41,785	1.27 (1.04–1.56)	1.27 (1.04–1.56)
Premenopausal TN with fluid, an abnormal exam, and a BMI < 30	158	41,838	0.95 (0.77–1.18)	0.0232 (0.77–1.18)
Premenopausal TN with fluid, an abnormal exam, and a BMI ≥ 30	47	41,949	0.28 (0.20–0.39)	0.28 (0.20–0.39)
Postmenopausal TN with fluid and a normal exam	862	41,134	5.19 (4.40–6.13)	5.28 (4.47–6.24)
Postmenopausal TN with fluid, a normal exam, and a BMI < 30	675	41,321	4.07 (3.43–4.82)	4.12 (3.47–4.88)
Postmenopausal TN with fluid, a normal exam, and a BMI ≥ 30	181	41,815	1.09 (0.88–1.35)	1.09 (0.88–1.35)
Postmenopausal TN with fluid and an abnormal exam	666	41,330	4.01 (3.39–4.75)	4.06 (3.42–4.82)
Postmenopausal TN with fluid, an abnormal exam, and a BMI < 30	512	41,484	3.08 (2.59–3.67)	3.11 (2.61–3.71)
Postmenopausal TN with fluid, an abnormal exam, and a BMI ≥ 30	149	41,847	0.90 (0.72–1.12)	0.90 (0.72–1.12)

Note: 95% CIs are in brackets. Body Mass Index (BMI)

**Table 6 diagnostics-12-00144-t006:** PRs and ORs comparing fluid status in positive TVS findings.

Group	Fluid-Positive	Fluid-Negative	PR (95% CI)	OR (95% CI)
Premenopausal TN with fluid, a normal exam, and a BMI < 30	166	41830	1	1
Benign findings (FPs)	31	614	12.16 (8.35–17.70)	12.72 (8.60–18.82)
Borderline tumor (low malignant potential (LMP))	2	26	18.07 (4.71–69.30)	19.38 (4.56–82.33)
Ovarian cancer (TP)	13	78	36.14 (21.36–61.14)	42.00 (22.90–77.03)
FN for ovarian cancer	4	21	40.48 (16.28–100.65)	48.00 (16.30–141.35)

Note: 95% CIs are in brackets. Body Mass Index (BMI).

## Data Availability

The data presented in this study are available on request from the corresponding author. The data are not publicly available due to the concern for infringement on research subjects’ privacy.

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
