# Peer review of "Significance of Pelvic Fluid Observed during Ovarian Cancer Screening with Transvaginal Sonogram"

_diagnostics, 2022, doi:10.3390/diagnostics12010144_

Round 1
Reviewer 1 Report
In this article, the authors examine the frequency and duration of free fluid during TVS exams in a large screening population and correlate the findings with the patient diagnosis. The primary objective was to determine if free fluid provides additional information in predicting ovarian malignancy compared to ultrasound alone. The manuscript is straightforward, well written, and concise and has clear results within the scope of a prospective analysis. Definitely deserves to be published and is a valuable contribution to the “diagnostics” journal. Some minor flaws need to be addressed before publication.
Minor points:
[1] “1. Introduction”, Lines 34-51:
The introduction section should include some epidemiological data of ovarian cancer (mortality, morbidity), the pathogenesis, a statement about the lack of screening methods, the genetic background (BRCA genes, homologous recombination), the therapeutic strategies, etc.
[2] General comment:
I would recommend the authors to incorporate an additional table, summarizing the 4 reported in the manuscript ovarian cancer screening trials utilizing transvaginal sonography (references 4-7). The variables that could be included are
-
Study design
-
Screening tests
-
Number screened/detected patients
-
Number of invasive cancers
-
Stage of the disease (I, II, III and IV)
-
Survival benefit (Yes vs No)
Author Response
Comments and Suggestions for Authors
Reviewer 1
In this article, the authors examine the frequency and duration of free fluid during TVS exams in a large screening population and correlate the findings with the patient diagnosis. The primary objective was to determine if free fluid provides additional information in predicting ovarian malignancy compared to ultrasound alone. The manuscript is straightforward, well written, and concise and has clear results within the scope of a prospective analysis. Definitely deserves to be published and is a valuable contribution to the “diagnostics” journal. Some minor flaws need to be addressed before publication.
Minor points:
[1] “1. Introduction”, Lines 34-51:
“The introduction section should include some epidemiological data of ovarian cancer (mortality, morbidity), the pathogenesis, a statement about the lack of screening methods, the genetic background (BRCA genes, homologous recombination), the therapeutic strategies, etc.”
In accordance with Reviewer #1, the manuscript has been revised to present epidemiological data of ovarian cancer (mortality, morbidity), the pathogenesis, a statement about screening methods, the genetic background (BRCA genes, homologous recombination), and therapeutic strategies. This modification ends at line 52 and is included below. A summary of screening effectiveness is added in Table 1.
Ovarian cancer continues to be the most lethal gynecological cancer with most patients facing a diagnosis of late stage metastatic disease that is associated with a 5-year survivals of only 30%. While the lifetime risk of ovarian cancer is less than breast cancer, it has a death-to-incidence ratio that is 3-4 times more than breast cancer [[1],[2],[3]]. Ovarian cancers can arise from the ovary as well as the fallopian tube [[4],[5],[6],[7],[8],[9],[10],[11]] and occur as five major histological subtypes (high grade serous, low-grade serous, endometrioid, clear cell and mucinous) that have been explored by immunochemical, genetic and homologous recombination approaches [[12]]. Deleterious mutations in DNA repair genes can drive defective homologous recombination and are emerging biomarkers of sensitivity/insensitivity to PARP inhibitors which interfere with the ability of poly-ADP ribose polymerase (PARP) to repair treatment-mediated DNA damage in cancer cells [[13]]. Despite technically advanced treatment strategies in precision medicine and immunotherapy, long term durable treatment responses have not yet been achievable [[14],[15]]. However, long-term survival rates that are greater than 90% in women with stage I ovarian cancer have focused an advocacy for screening efforts to detect early stage disease [[16]].
[2] General comment:
I would recommend the authors to incorporate an additional table, summarizing the 4 reported in the manuscript ovarian cancer screening trials utilizing transvaginal sonography (references 4-7). The variables that could be included are
- Study design
- Screening tests
- Number screened/detected patients
- Number of invasive cancers
- Stage of the disease (I, II, III and IV)
Survival benefit (Yes vs No)
In accordance with Reviewer #1 this summary has been added as Table 1, as shown below and occurring at line 154 with text edits lines 56-60.
Table 1. Summary of Ovarian Screening Trials
|
Study |
KYOVS |
PLCO |
SCSOCS |
UKCTOCS |
|||||
|
Study Design |
Prospective Cohort (Ongoing) |
Intent to Treat RCT (Closed) |
Intent to Treat RCT (Closed) |
Intent to Treat RCT (Closed) |
|
||||
|
Number Screened |
48,925 a |
34,253b 34,304* |
41,688b 40,799* |
50,625c 50,623a 101,314* |
|
||||
|
Total Screens |
326,998 |
150,598 |
156,747 |
345,570c 327,775a |
|
||||
|
Invasive Ovarian Cancers Detected |
78 |
212 b |
27 |
522c 517a 1016* |
|
||||
|
Shift to Early Stage Diseased |
Yes (63%) |
No |
Yes (67%) |
Yes (39.2%)
|
|
||||
|
Survival Benefit |
Yes |
No |
No |
Yes
|
|
||||
aUSS alone, bUSS alone followed by Ca125, cCa125 followed by USS, *control, dStage I & II
Reviewer 2 Report
In the present manuscript, “Significance of Pelvic Fluid Observed During Ovarian Cancer Screening with Transvaginal Sonogram,” the authors investigated the significance of pelvic fluid in ovarian cancer screening using a large-scale dataset. The purpose of the study is clinically interesting, and this study potentially has a significant value. However, the result of the investigation feels strange and difficult to understand despite the simple concept. One of the reasons is that the study design is inappropriate to investigate the purpose of the study. In the introduction section, the authors mentioned that the purpose of the study is to determine if free fluid provides additional information in predicting ovarian malignancy compared to ultrasound alone. However, the authors classified the patients based on the final diagnosis. I think that the patients should be stratified with ultrasound findings; for example, those with ovarian tumor with fluid, those with ovarian tumor without fluid, those with no ovarian mass but fluid, and those without any finding. Therefore, I think the authors should re-analyze the dataset, which will improve the impact of the study.
Author Response
- In the present manuscript, “Significance of Pelvic Fluid Observed During Ovarian Cancer Screening with Transvaginal Sonogram,” the authors investigated the significance of pelvic fluid in ovarian cancer screening using a large-scale dataset. The purpose of the study is clinically interesting, and this study potentially has a significant value. However, the result of the investigation feels strange and difficult to understand despite the simple concept. One of the reasons is that the study design is inappropriate to investigate the purpose of the study. In the introduction section, the authors mentioned that the purpose of the study is to determine if free fluid provides additional information in predicting ovarian malignancy compared to ultrasound alone. However, the authors classified the patients based on the final diagnosis I think that the patients should be stratified with ultrasound findings; for example, those with ovarian tumor with fluid, those with ovarian tumor without fluid, those with no ovarian mass but fluid, and those without any finding. Therefore, I think the authors should re-analyze the dataset, which will improve the impact of the study.
In accordance with reviewer #2, we re-analyzed the data with added information on sonographic findings in lines 133-145. In particular, information has been added on fluid in TVS exams prior to the final diagnosis.
Since 1987, the University of Kentucky Ovarian Screening Program has performed 326,998 TVS screens on 48,925 women. We observed free fluid in 2,001 (4.1%) of those encounters. True positive screens included 78 ovarian malignancies (13 fluid-positive, 16.7%), 20 tumors of low malignant potential (two fluid-positive, 10%), and 23 malignancies of non-ovarian origin (three fluid-positive, 13%). Only one of the true positive cases was observed to have fluid present in a prior normal TVS exam. There were 614 FP screens classified as high risk for ovarian cancer but found to have benign pathology (31 fluid-positive, 5%). Nine of these cases occurred on a prior normal TVS exam. The TN screens included 41,996 cases that screened negative for malignancy and did not develop ovarian carcinoma (1,948 fluid-positive, 4.6%: 1,071 fluid positive cases were associated with a normal TVS exam while 877 were associated with an abnormal TVS exam). There were 21 cases of ovarian cancer diagnosed within 12 months of a FN TVS scan (4 fluid-positive, 19%). All of these cases occurred in the absence of an abnormal TVS exam.
[1]. Howlader, N.; Noone, A.M.; Krapcho, M.; Garshell, J.; Miller, D.; Altekruse, S.F.; Kosary, C.L.; Yu, M.; Ruhl, J.; Tatalovich, Z.; et al. Seer Cancer Statistics Review; National Cancer Institute: Bethesda, MD, USA, 2015; pp. 1975–2016. Available online: https://seer.cancer.gov/csr/1975_2016/ (accessed on 14 December 2021).
[1]. Pavlik, E.J.; van Nagell, J.R., Jr. Early detection of ovarian tumors using ultrasound. Womens Health Lond 2013, 9, 39–55. [Google Scholar] [CrossRef]
[1]. Committee on the State of the Science in Ovarian Cancer Research; Board on Health Care Services; Institute of Medicine; National Academies of Sciences, Engineering, and Medicine. Ovarian Cancers: Evolving Paradigms in Research and Care; National Academies Press: Washington, DC, USA, 2016. Available online: https://www.ncbi.nlm.nih.gov/books/NBK367618/ (accessed on 14 December 2021). [CrossRef]
[1]. Kurman, R.J.; Shih, I.-M. Molecular pathogenesis and extraovarian origin of epithelial ovarian cancer—Shifting the paradigm. Hum. Pathol. 2001, 42, 918–931.
[1]. Shih, I.-M.; Kurman, R.J. Ovarian tumorigenesis: A proposed model based on morphological and molecular genetic analysis. Am. J. Pathol. 2004, 164, 1511–1518.
[1]. Crum, C.P. Intercepting pelvic cancer in the distal fallopian tube: Theories and realities. Mol. Oncol. 2009, 3, 165–170.
[1]. Gershenson, D.M.; Tortolero-Luna, G.; Malpica, A.; Baker, V.V.; Whittaker, L.; Johnson, E.; Follen, M.M. Ovarian intraepithelial neoplasia and ovarian cancer. Obstet. Gynecol. Clin. N. Am. 1996, 23, 475–543.
[1]. Kindelberger, D.W.; Lee, Y.; Miron, A.; Hirsch, M.S.; Feltmate, C.; Medeiros, F.; Callahan, M.J.; Garner, E.O.; Gordon, R.W.; Birch, C.; et al. Intraepithelial carcinoma of the fimbria and pelvic serous carcinoma: Evidence for a causal relationship. Am. J. Surg. Pathol. 2007, 31, 161–169.
[1]. Crum, C.P.; McKeon, F.D.; Xian, W. The oviduct and ovarian cancer: Causality, clinical implications, and “targeted prevention”. Clin. Obstet. Gynecol. 2012, 55, 24–35.
[1]. Soong, T.R.; Howitt, B.E.; Horowitz, N.; Nucci, M.R.; Crum, C.P. The fallopian tube, “precursor escape” and narrowing the knowledge gap to the origins of high-grade serous carcinoma. Gynecol. Oncol. 2019, 152, 426–433.
[1]. Menon U, Karpinskyjj C, Gentry-Maharaj A. Ovarian cancer prevention and screening. Obstet Gynecol 2018;131:909–27.
[1]. Barnes, B.M., Nelson, L., Tighe, A. et al. Distinct transcriptional programs stratify ovarian cancer cell lines into the five major histological subtypes. Genome Med 13, 140 (2021). https://doi.org/10.1186/s13073-021-00952-5
[1]. Ksenija Nesic, Olga Kondrashova, Rachel M. Hurley, Cordelia D. McGehee, Cassandra J. Vandenberg, Gwo-Yaw Ho, Elizabeth Lieschke, Genevieve Dall, Nirashaa Bound, Kristy Shield-Artin, Marc Radke, Ashan Musafer, Zi Qing Chai, Mohammad Reza Eftekhariyan Ghamsari, Maria I. Harrell, Damien Kee, Inger Olesen, Orla McNally, Nadia Traficante, Australian Ovarian Cancer Study, Anna DeFazio, David D.L. Bowtell, Elizabeth M. Swisher, S. John Weroha, Katia Nones, Nicola Waddell, Scott H. Kaufmann, Alexander Dobrovic, Matthew J. Wakefield and Clare L. Scott. Acquired RAD51C Promoter Methylation Loss Causes PARP Inhibitor Resistance in High-Grade Serous Ovarian Carcinoma.
Cancer Res September 15 2021 (81) (18) 4709-4722; DOI: 10.1158/0008-5472.CAN-21-0774
[1]. Immune Checkpoint Inhibitors in Ovarian Cancer: Can We Bridge the Gap Between IMagynation and Reality? Panagiotis A. Konstantinopoulos and Stephen A. Cannistra. Journal of Clinical Oncology 2021 39:17, 1833-1838
[1]. Rebecca L. Porter and Ursula A. Matulonis. Checkpoint Blockade: Not Yet NINJA Status in Ovarian Cancer. Journal of Clinical Oncology 2021 39:33, 3651-3655
[1]. Shengqing Gu, Stephanie Lheureux, Azin Sayad, Paulina Cybulska, Liat Hogen, Iryna Vyarvelska,
Dongsheng Tu, Wendy R. Parulekar, Matthew Nankivell, Sean Kehoe, Dennis S. Chi, Douglas A. Levine, Marcus Q. Bernardini, Barry Rosen, Amit Oza, Myles Brown, Benjamin G. Neel. Computational modeling of ovarian cancer dynamics suggests optimal strategies for therapy and screening.
Proceedings of the National Academy of Sciences Jun 2021, 118 (25) e2026663118; DOI: 10.1073/pnas.2026663118
Round 2
Reviewer 2 Report
I think the manuscript has improved.
Author Response
We appreciate the reviewers’ comments.